# Multiple drivers of the COVID-19 spread: The roles of climate, international mobility, and region-specific conditions

Yasuhiro Kubota[1,2]*, Takayuki Shiono[1], Buntarou Kusumoto[3], Junichi Fujinuma[1]

**1** Faculty of Science, University of the Ryukyus, Okinawa, Japan, **2** Think Nature Inc., Okinawa, Japan, **3** University Museum, University of the Ryukyus, Okinawa, Japan

* Kubota.yasuhiro@gmail.com

**Data Availability Statement:** All relevant data are within the paper and its Supporting Information files.

## Abstract

Following its initial appearance in December 2019, coronavirus disease 2019 (COVID-19) quickly spread around the globe. Here, we evaluated the role of climate (temperature and precipitation), region-specific COVID-19 susceptibility (BCG vaccination factors, malaria incidence, and percentage of the population aged over 65 years), and human mobility (relative amounts of international visitors) in shaping the geographical patterns of COVID-19 case numbers across 1,020 countries/regions, and examined the sequential shift that occurred from December 2019 to June 30, 2020 in multiple drivers of the cumulative number of COVID-19 cases. Our regression model adequately explains the cumulative COVID-19 case numbers (per 1 million population). As the COVID-19 spread progressed, the explanatory power ($R^2$) of the model increased, reaching > 70% in April 2020. Climate, host mobility, and host susceptibility to COVID-19 largely explained the variance among COVID-19 case numbers across locations; the relative importance of host mobility and that of host susceptibility to COVID-19 were both greater than that of climate. Notably, the relative importance of these factors changed over time; the number of days from outbreak onset drove COVID-19 spread in the early stage, then human mobility accelerated the pandemic, and lastly climate (temperature) propelled the phase following disease expansion. Our findings demonstrate that the COVID-19 pandemic is deterministically driven by climate suitability, cross-border human mobility, and region-specific COVID-19 susceptibility. The identification of these multiple drivers of the COVID-19 outbreak trajectory, based on mapping the spread of COVID-19, will contribute to a better understanding of the COVID-19 disease transmission risk and inform long-term preventative measures against this disease.

## Introduction

The spread of infectious diseases through host–pathogen interaction is fundamentally underpinned by macroecological and biogeographical processes [1, 2]; key processes include virus origination, dispersal, and evolutionary diversification through local transmissions in human societies [3]. Since December 2019, coronavirus disease 2019 (COVID-19), caused by sudden

**Funding:** This study is not supported by any research funds.

**Competing interests:** The authors have declared that no competing interests exist.

acute respiratory syndrome coronavirus 2 (SARS-CoV-2), has quickly spread worldwide from Wuhan, China [4]. The disease transmission geography of COVID-19 was highly heterogeneous; some countries (e.g., Japan) had cases from the earliest stage of this pandemic, but their increase in the number of new cases was relatively moderate, whereas others (e.g., EU nations and the USA) experienced later but substantial COVID-19 outbreaks. To predict infection risk on the global scale, the forces driving the COVID-19 outbreak patterns must be identified [5]. Additionally, capturing region-specific factors influencing the outbreak progress is critically important for improving long-term control measures against this ongoing pandemic.

Infectious diseases due to respiratory viruses are empirically characterized by a seasonal nature [6]. Moriyama et al. [7] described a framework to better understand the mechanisms of virus transmission; air temperature, absolute/relative humidity, and sunlight are jointly associated with virus viability/stability and host defense, and thereby human-to-human transmission of COVID-19 is promoted by contact rates along with host susceptibility (or immunity) to COVID-19. From this viewpoint, several research groups have focused on relevant factors separately and quickly examined the role of climate [8–10], international mobility linked to human contact [11, 12], and community-based host susceptibility to COVID-19 [13]. However, these analyses were inconclusive, and the relative importance of these factors in promoting the disease expansion of COVID-19 remains unclear.

This study assessed multiple potential drivers of the COVID-19 spread, by conducting an analysis of time-series data on the number of confirmed COVID-19 cases from December 2019 through June 2020, as well as on country/region-specific variables, e.g. socioeconomic conditions and screening effort (number of SARS-CoV-2 PCR tests conducted), that could potentially affect the number of COVID-19 cases. Specifically, we explored the roles of climate, international mobility, and region-specific conditions in the disease expansion by controlling covariates. In this analysis, we evaluated the relative importance of climate (temperature and precipitation relevant to habitat suitability for SARS-CoV-2), region-specific COVID-19 susceptibility (BCG vaccination factors, malaria incidence, and the relative proportion of citizens aged over 65 years in the population, as these were hypothesized to be linked with host susceptibility to COVID-19), and human mobility (international travel) in shaping the current geographical patterns of COVID-19 spread around the world.

## Materials and methods

### Data sources

We compiled geographic data on the number of reported COVID-19 cases per day from December 2019 to June 30, 2020. We collected the numbers of COVID-19 cases for 1,020 countries/regions from various sources (see S1 Appendix for a list of data sources for the COVID-19 cases). We then calculated the length of time (in days) since the onset of COVID-19 spread as defined by the date of the first confirmed case in each country or region. We also examined the number of SARS-CoV-2 PCR tests conducted based on data published by the World Health Organization (WHO) (https://ourworldindata.org/covid-testing) to assess the influence of sampling effort on the number of confirmed cases of COVID-19.

For each country or region, we compiled several environmental variables. For mapping cases of COVID-19, the longitude and latitude of the largest city and area for each country or region were extracted from GADM maps and data (https://gadm.org/index.html). Based on the geocoordinates of the cities, we collected the climatic data of mean precipitation (mm month$^{-1}$) and temperature (˚C) from January to June (WorldClim) using WorldClim version 2.1 climate data (https://www.worldclim.org/data/worldclim21.html) at a resolution of 2.5 arc-minutes grid cells that contained a country or region.

Regarding international travel linked to the disease transmission, we compiled the average annual number of foreign visitors (per year) for individual countries/regions from data published by the World Tourism Organization (https://www.e-unwto.org/toc/unwtotfb/current). We then calculated the relative amount of foreign visitors per population of each country or region to use in the analysis.

Regarding region-specific host susceptibility to COVID-19, we collected data on the following three epidemiologic properties: the proportion of the population aged over 65 years, the malaria incidence (per year), and information regarding bacillus Calmette–Guérin (BCG) vaccination. We included these attributes in our analyses based on the assumptions that BCG vaccination and/or recurrent treatment with anti-malarial medications could be associated with providing some protection against COVID-19 [13, 14]. We compiled BCG data from the WHO (https://www.who.int/malaria/data/en/) and (https://apps.who.int/gho/data/view.main.80500?lang=en) and the BCG Atlas Team (http://www.bcgatlas.org/) on the following five attributes: i) the number of years since BCG vaccination was started (BCG_year); ii) the present situation regarding BCG vaccination (BCG_type), split into all vaccinated, partly vaccinated, vaccinated once in the past, or never vaccinated; iii) the relative frequency of post-1980 (i.e., the past 40 years) BCG vaccination for people aged less than 1 year old (BCG_rate); iv) the number of BCG vaccinations (MultipleBCG), describing countries as never having vaccinated their citizens with BCG, vaccinated their citizens with BCG only once, vaccinated their citizens with BCG multiple times in the past, or currently vaccinate their citizens with BCG multiple times; and v) tuberculosis cases per 1 million people (TB). These BCG-related variables are strongly intercorrelated. Therefore, we reduced the dimensions of these variables (BCG_year, BCG_type, BCG_rate, MultipleBCG, and TB) by extracting the first axis of the PCA analysis: the score of the PCA 1 axis was negatively correlated with the five variables, so the PCA 1 score multiplied by –1 was defined as the BCG vaccination effect.

We also compiled socioeconomic data for each country or region. The population size, population density (per $km^2$) (Gridded Population of the World GPW, v4.; https://sedac.ciesin.columbia.edu/data/collection/gpw-v4), gross domestic product (GDP in US dollars), and GDP per person were obtained from national census data (World Development Indicators; https://datacatalog.worldbank.org/dataset/world-development-indicators).

## Statistical analyses

The monthly pattern for the cumulative number of COVID-19 cases in each country/region was visualized in relation to the geography, biome type, and climate (mean temperature and annual precipitation) of that location. In addition, the pattern of increasing COVID-19 case numbers was evaluated based on country type, with individual countries being classified into four types defined by the number of COVID-19 cases per week and the date of outbreak onset.

To ensure the robustness of our results, we investigated the relationship between various environmental variables (climate, host susceptibility to COVID-19, international human mobility, and socioeconomic factors) and the number of COVID-19 cases (per 1 million population) using the two different approaches: conventional multiple linear regression and random forest, which is a machine-learning model [15]. We separately modeled the cumulative number of COVID-19 cases (per 1 million population) in successive periods from December 2019 to June 30, 2020.

In the multiple regression analysis, we set the log-scaled cumulative number of COVID-19 cases within a period as the response variable and the climatic factors (mean temperature, squared mean temperature, and log-scaled monthly precipitation), socioeconomic conditions (log-scaled population density and GDP per person), international human mobility (the

relative amount of foreign visitors per population) and region-specific COVID-19 susceptibility (the percentage of people aged $\geq$ 65 years, the log-scaled relative incidence of malaria, and the BCG vaccination effect) as explanatory variables.

To control for country/region-specific observation biases, we included the length of time (measured in days) since the first confirmed COVID-19 case in each country/region and the number of COVID-19 tests conducted (as a measure of sampling effort) as covariates. In addition, we applied the trend surface method to take spatial autocorrelation into account as a covariate; we added the first eigenvector of the geo-distance matrix among the countries or regions, which was computed using the geocoordinates of the largest city, as a covariate [16]. The explanatory power of the model was evaluated by the adjusted coefficient of determination ($R^2$). We also calculated the relative importance of each explanatory variable in a regression model according to its partial coefficient of determination and determined the predominant variables that explained the variance in the response variables. The statistical significance of each variable was determined by conducting F-test. All the explanatory variables were standardized to have a mean of zero and a variance of one before these analyses. The explanatory factors of the regression model were compared between the four country types.

In the random forest model, we used the same set of response and explanatory variables, as well as the same covariates. In each run of the random forest analysis, we generated 1,000 regression trees. The model performance was evaluated by the proportion of variance explained by the model. We evaluated the relative importance of each explanatory variable based on the increase in the mean squared error when the variable was permutated.

Before these analyses, we tested the collinearity between the explanatory variables by calculating the variance inflation factor (VIF). For the study period, the largest VIF value was 8.56, and the VIF at June 30, 2020 was 8.56, indicating the absence of multicollinearity in the regression.

To confirm the testing effort bias on the number of confirmed cases, we conducted an additional analysis that accounted for the number of conducted tests (i.e., sampling efforts) in individual countries/regions, as a covariate in the model. Note that this analysis was applied to the data from 128/828 countries/regions, because testing data for many countries is currently unavailable (https://ourworldindata.org/covid-testing).

All analyses were performed with the R environment for statistical computing [17]; the 'sf' package was used for graphics artworks [18] and the 'randomForest' package was used for the random forest analysis [19].

## Results and discussion

COVID-19 (as measured by the number of cases per 1 million population) spread rapidly across the globe after it first appeared in Wuhan, China in December, 2019 (Li et al. 2020) (Fig 1; S1 Video), but the outbreak appears to have occurred in particular climates around 8˚C and 26˚C or biomes (Fig 2; S2 Video). Moreover, the patterns of increasing number of COVID-19 cases per week varied among the countries that are characterized by different COVID-19 spread dates (Fig 3 and S1 Fig).

Although the COVID-19 case numbers may not be suitable for conducting epidemiological analyses, such as modelling the disease growth dynamics, the available COVID-19 case data can be still informative for the implementation of containment and/or suppression measures because the number of the confirmed cases is directly linked to the consumption of medical resources for combatting the COVID-19 pandemic. Here, we observed that the cumulative number of the COVID-19 cases (per 1 million population) according to the disease spread progression was significantly correlated with variables related to climate, international human

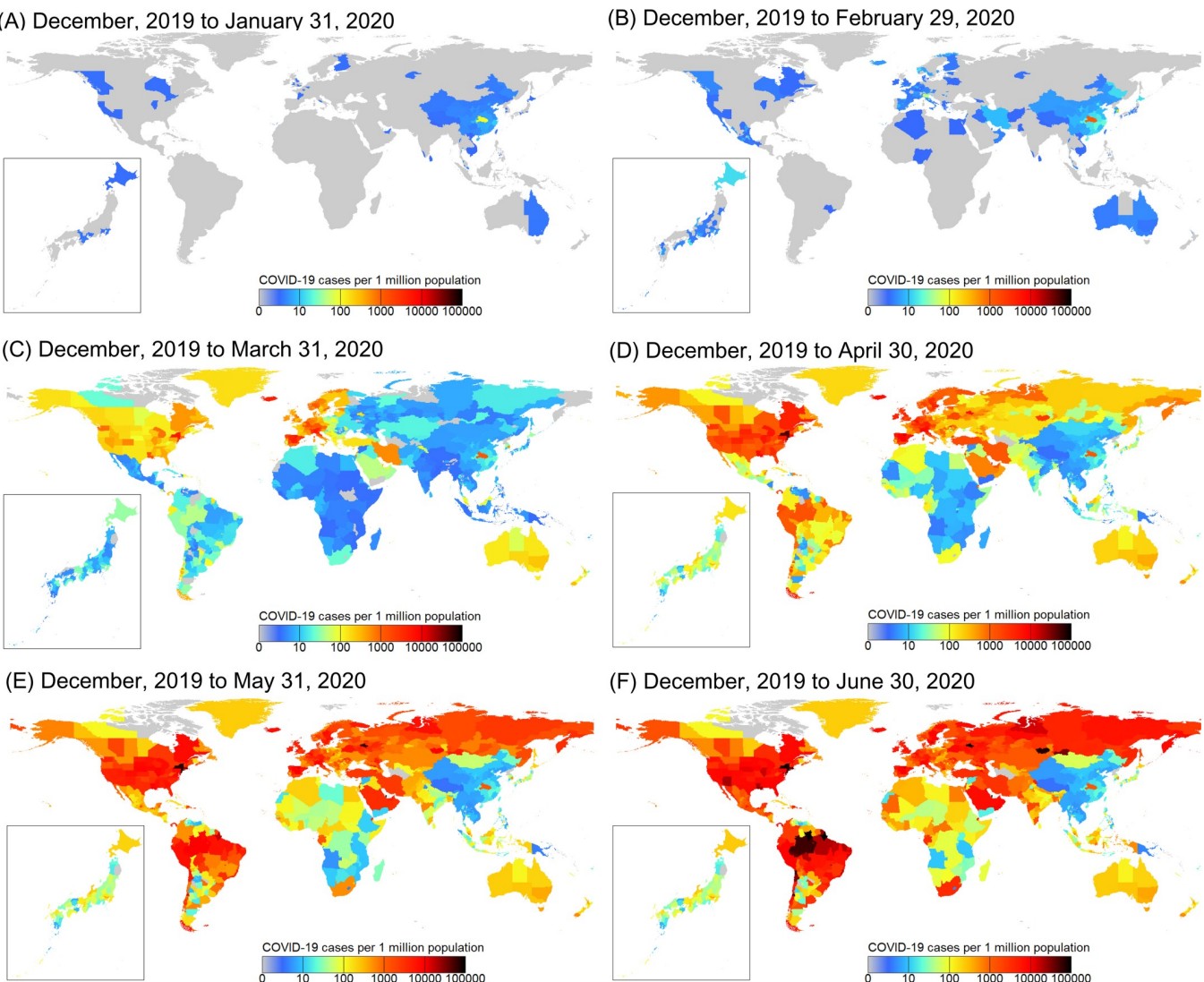

**Fig 1. Geographical distribution of COVID-19 cases (per 1 million population) for 1,020 countries/regions worldwide.** (A–F) Monthly patterns for the cumulative number of COVID-19 cases on January 31, 2020 (A), February 29, 2020 (B), March 31, 2020 (C), April 30, 2020 (D), May 31, 2020 (E), and June 30, 2020 (F) based on the cumulative number of day-to-day COVID-19 cases since December 2019. See S3 Video. The map was prepared using shapefile reprinted from a freely available database (GADM; www.gadm.org).

mobility, and host susceptibility to COVID-19, at successive periods since December, 2019 (Fig 4).

The explanatory power, i.e., coefficient of determination ($R^2$), of the model as the COVID-19 pandemic progressed, reaching >70% in April 2020 (Fig 5A). The number of days from case onset had some explanatory power (> 20%) in January, 2020, but this factor quickly lost its influence as the pandemic progressed (Fig 5B). As the influence of this factor waned, other variables (related to climate, human mobility, and host susceptibility to COVID-19) exhibited the increasing explanatory powers (Fig 5B). After April 2020, the explanatory power of variables related to human mobility and host susceptibility to COVID-19 rapidly decreased. After this, the explanatory power of human population and climate factors increased. These results demonstrate that the impact of virus dispersability between/within regions was predominant in the beginning stage of the pandemic (Fig 5).

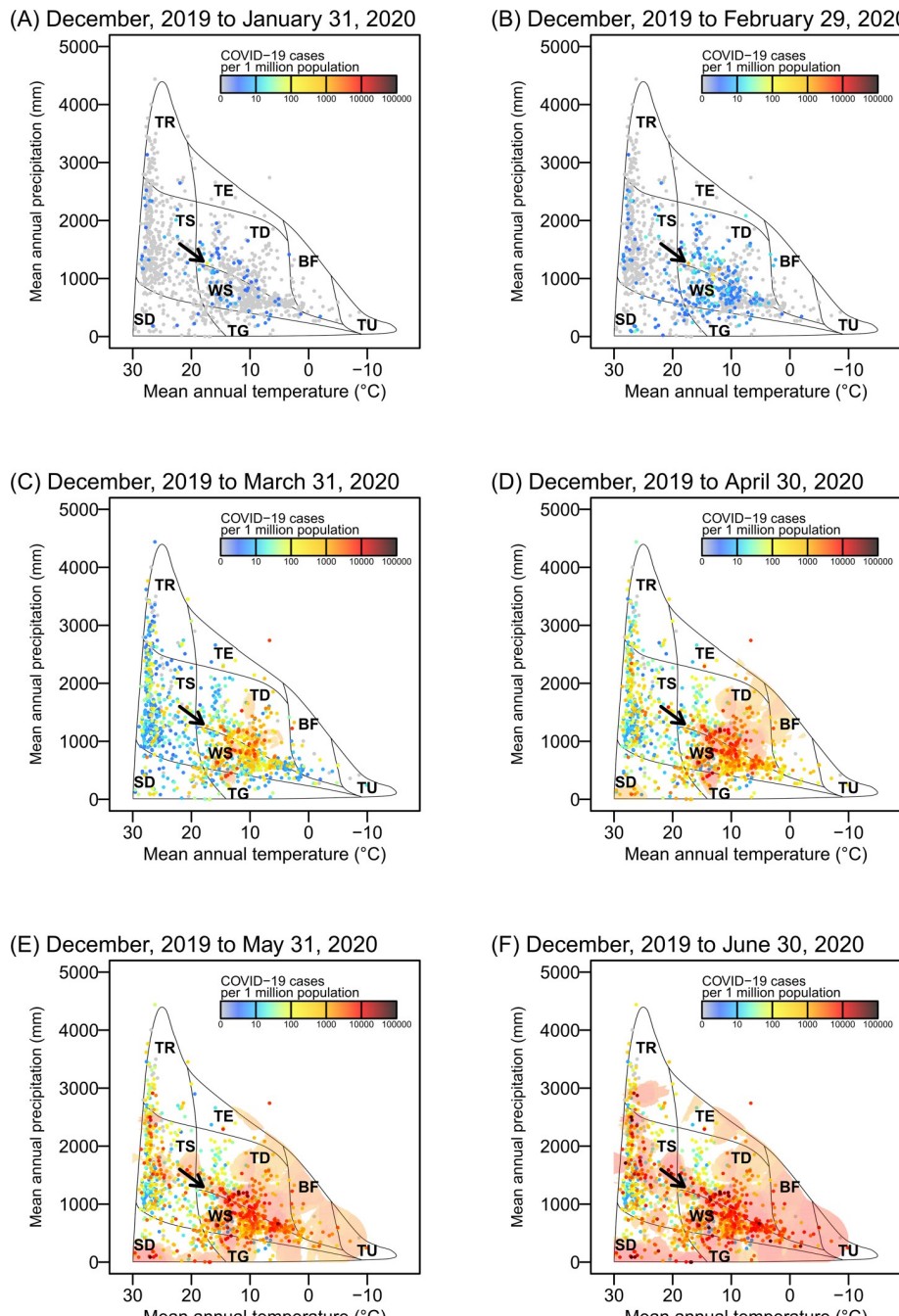

**Fig 2. The distribution of COVID-19 cases across biome types based on the relationship between mean temperature and annual precipitation.** Biome classification is based on the scheme by Whittaker [20]. (TR) tropical rain forest; (TS) tropical seasonal forest/savanna; (TE) temperate rain forest; (SD) subtropical desert; (TD) temperate deciduous forest; (WS) woodland/shrubland; (TG) temperate grassland/desert; (BF) boreal forest, (TU) tundra. Colors indicate the number of COVID-19 cases (per 1 million population) and also contours of climatic regions with ≥1000 cases per 1 million population. (A–F) Monthly patterns for the cumulative number of COVID-19 cases on January 31, 2020 (A), February 29, 2020 (B), March 31, 2020 (C), April 30, 2020 (D), May 31, 2020 (E), and June 30, 2020 (F) based on the cumulative number of day-to-day COVID-19 cases since December 2019. Arrows indicate the location of Wuhan in China. See S4 Video.

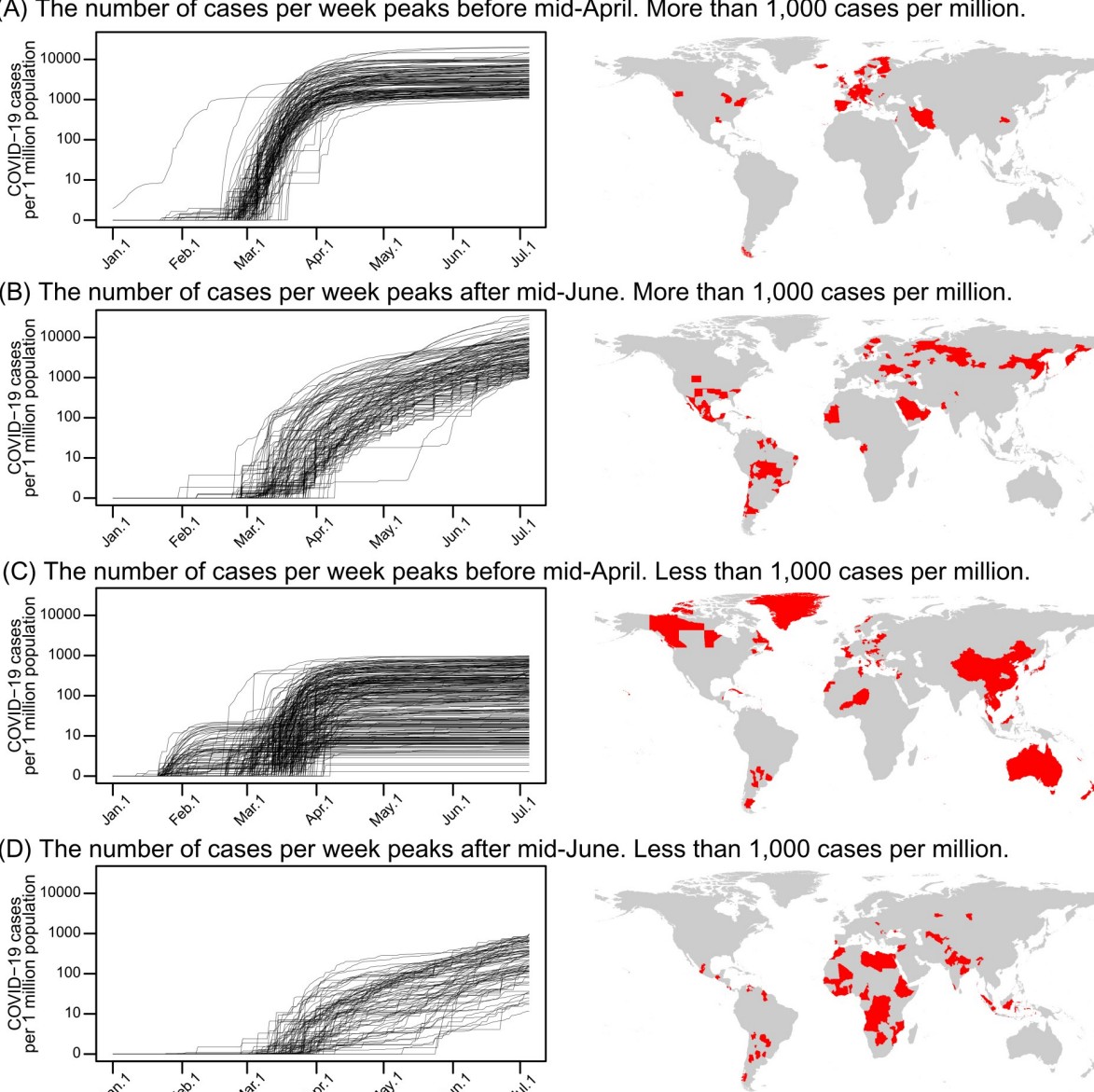

(A) The number of cases per week peaks before mid-April. More than 1,000 cases per million.

(B) The number of cases per week peaks after mid-June. More than 1,000 cases per million.

(C) The number of cases per week peaks before mid-April. Less than 1,000 cases per million.

(D) The number of cases per week peaks after mid-June. Less than 1,000 cases per million.

**Fig 3. Patterns for the cumulative number of COVID-19 cases (per 1 million population) in relation to country type.** Based on the pattern of increasing COFVID-19 case numbers, individual countries were classified into four types (A–D): (A) Type A, countries that had a peak in the number of COVID-19 cases per week before the middle of April and had more than 1,000 COVID-19 cases per 1 million population; (B) type B, countries that exhibited an increase in the number of COVID-19 cases per week after the middle of June and had more than 1,000 COVID-19 cases per 1 million population; (C) type C, countries that had a peak in the number of COVID-19 cases per week before the middle of April and had less than 1,000 COVID-19 cases per 1 million population; (D) type D, countries that exhibited an increase in the number of COVID-19 cases per week after the middle of June and had less than 1,000 COVID-19 cases per 1 million population. The map was prepared using shapefile reprinted from a freely available database (GADM; www.gadm.org).

The standardized regression coefficients of the model greatly changed (from non-significant to significant) over the period from December, 2019 to April 12, 2020 (Fig 6). After February, 2020, the mean temperature was negatively correlated with the cumulative number of COVID-19 cases, whereas the mean precipitation was positively correlated with these values (Fig 6A–6C). After March, 2020, relative amount of foreign visitors per population and GDP per person were predominantly positively correlated with the cumulative number of COVID-

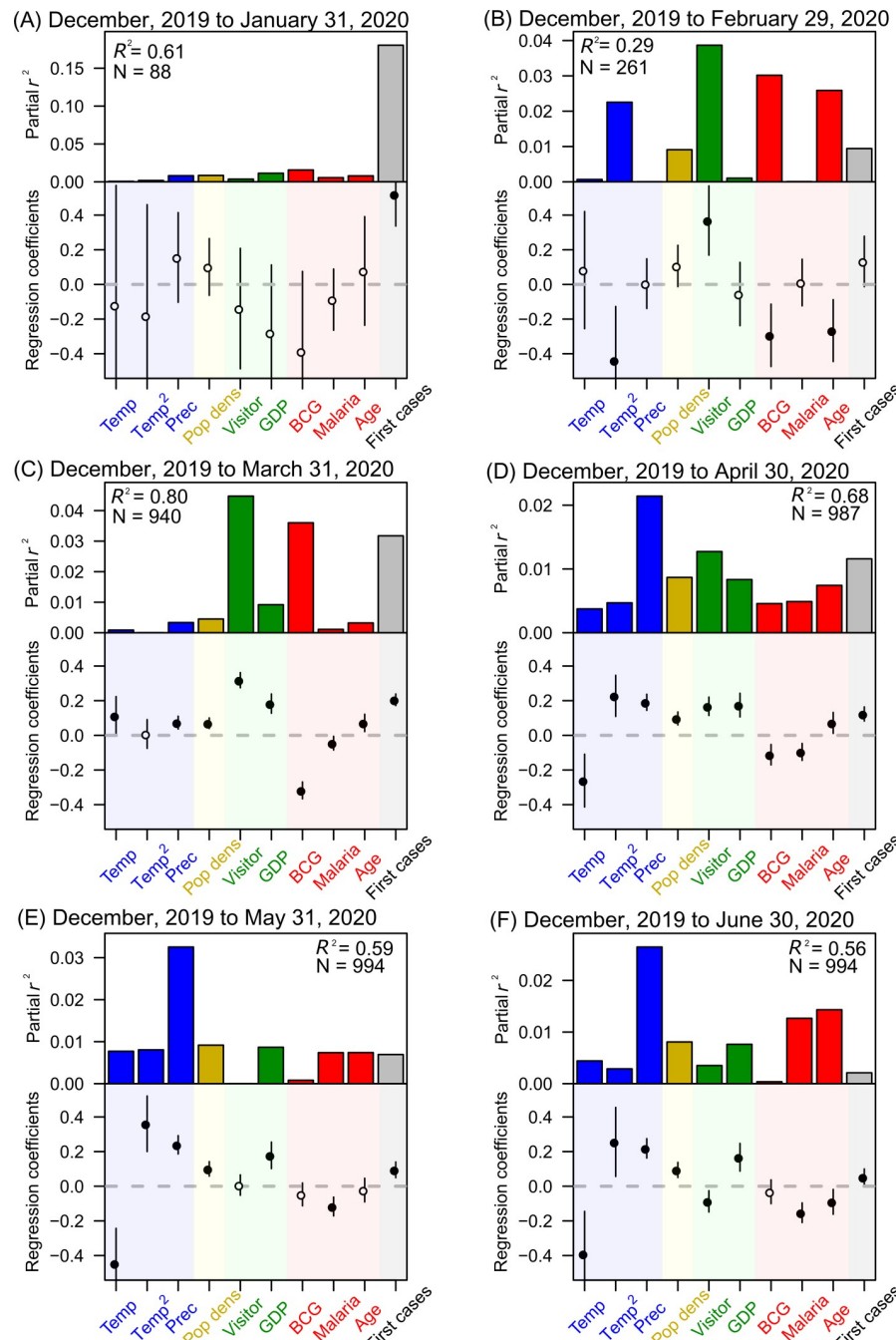

**Fig 4. Standardized regression coefficients and the partial coefficient of determination ($r^2$) of each explanatory factor in the regression model explaining the cumulative number of COVID-19 cases (per 1 million population).** (A–F) Values for the period from December 2019 to January 31, 2020 (A), February 29, 2020 (B), March 31, 2020 (C), April 30, 2020 (D), May 31, 2020 (E), or June 30, 2020 (F). Temp, mean temperature; Temp$^2$, squared mean temperature; Prec, mean monthly precipitation; Pop dens, population density; Visitor, relative amount of foreign visitors per population; GDP, gross domestic product per person; BCG, BCG vaccination effect as defined by the first PCA axis summarizing five variables related to BCG vaccination (see the Methods section for details); Malaria, relative malaria incidence; Age, relative proportion of the population aged ≥65 years; First cases, number of days from case onset. The regressions were conducted using ordinary least squares analyses. Vertical lines represent the 95% confidence intervals of parameters. Closed symbols indicate the significance of explanatory variables ($p < 0.05$). The coefficient of determination ($R^2$) for the overall model is also shown. A nonlinear modeling analysis was also conducted using the random forest method with the same set of response and explanatory variables and the same covariates; the results of this parallel analysis are shown in S2 Fig.

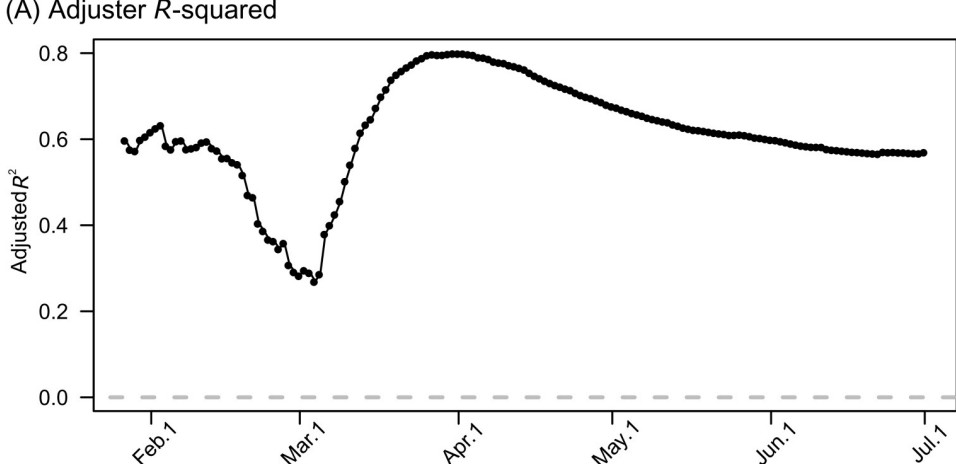

(A) Adjuster *R*-squared

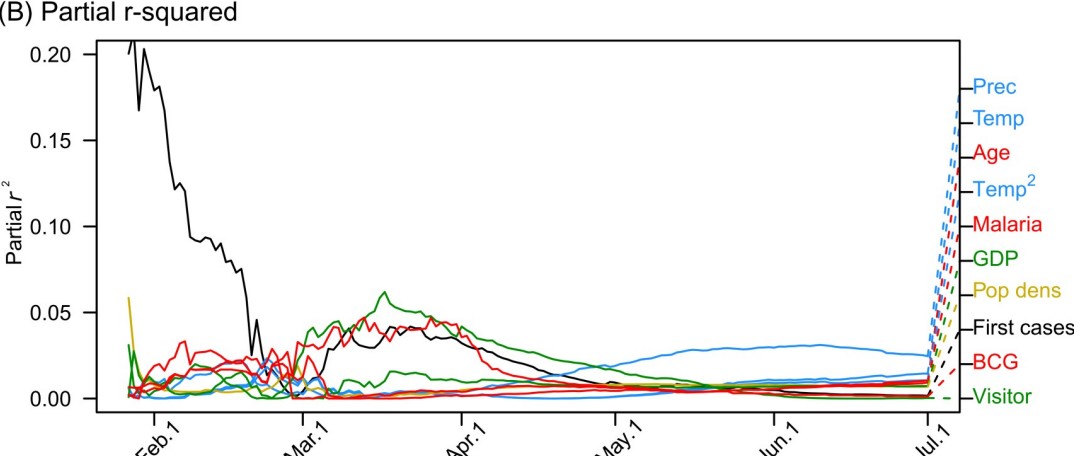

(B) Partial r-squared

**Fig 5. Coefficients of determination (adjusted $R^2$) of the regression model explaining the cumulative number of COVID-19 cases (per 1 million population) from December, 2019 to June 30, 2020.** (A) Overall coefficient of determination of the regression model; (B) coefficient of partial determination ($r^2$) for each explanatory variable in the model. The results shown are based on data starting from January, 2020, because the number of cases in December 2019 was insufficient for this analysis.

19 cases (Fig 6E and 6F). In contrast, since February or March 2020, the BCG vaccination factors and malaria incidence were consistently negatively correlated with the cumulative number of COVID-19 cases (Fig 6G and 6H). Population density was slightly positively correlated with the cumulative number of COVID-19 cases (Fig 6D). The relative proportion of the population aged ≥65 years was also positively correlated with these values, except for a temporary period where it was negatively correlated (Fig 6I). This shift from positive to negative correlation reflects the initial spread of COVID-19 in developed countries with relatively older population and the later (after May 2020) spread of COVID-19 in developing countries with relatively younger populations. In the early stage of COVID-19 spread, the number of days from case onset was strongly positively correlated with the cumulative number of COVID-19 cases (Fig 6J).

The results of the random forest model were generally consistent with those of the linear multiple regression model (S2 Fig). The relative importance of the variables related to human mobility and host susceptibility to COVID-19 (elderly population, BCG vaccination effect, and

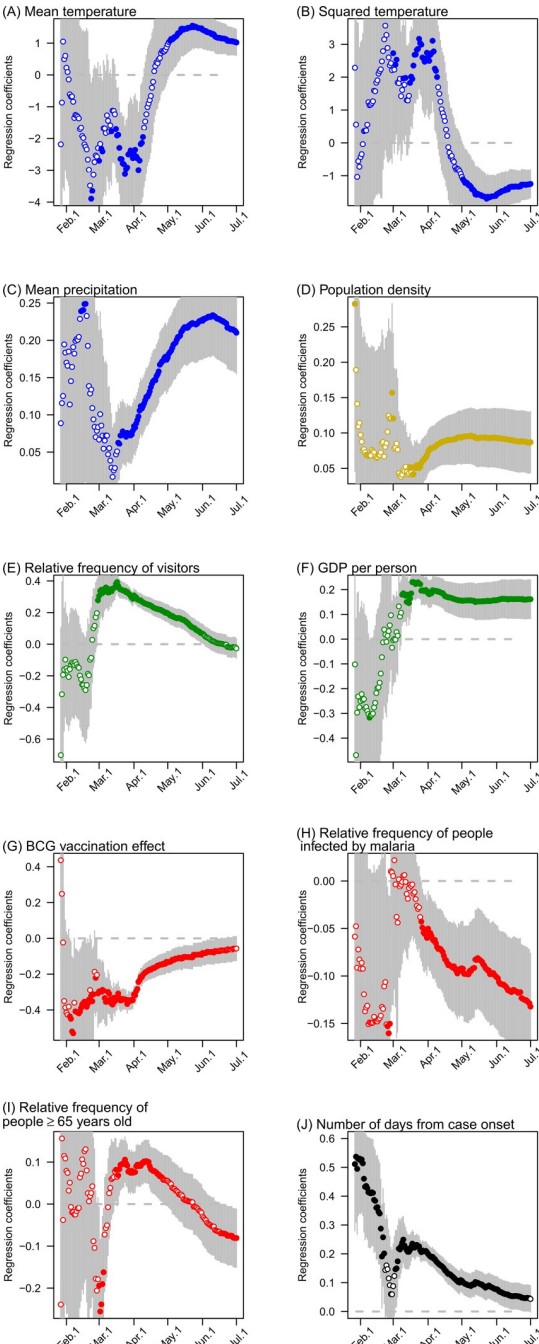

**Fig 6. Time-series pattern of the standardized regression coefficients of the model explaining the cumulative number of COVID-19 cases (per 1 million population) from December 2019 to June 30, 2020.** Vertical lines represent the 95% confidence intervals of parameters. The results are based on data starting from January 2020 because the number of COVID-19 cases in December 2019 was insufficient for this analysis.

malaria incidence) became predominant over time, whereas the relative importance of population density and the number of days from case onset decreased after March 2020. Moreover, additional analyses, which included the number of conducted COVID-19 tests as a covariate, revealed very similar patterns of regression coefficients, and their explanatory power (S3 Fig),

i.e., the roles of climate, international human mobility, and host susceptibility to COVID-19, became more pronounced as the pandemic progressed. Therefore, the nonlinearity of epidemic and region-specific testing bias had no serious influence on identifying the environmental drivers shaping the present COVID-19 distribution.

This study generally supports the findings of several recent reports, which found that climate [8–10], international human mobility [11, 12], and community-based host susceptibility to COVID-19 [13] jointly contributed to the spread of COVID-19. Notably, the explanatory power of these drivers substantially increased as the pandemic progressed, indicating a deterministic expansion of COVID-19 around the world.

Cross-border human mobility, which has been facilitated by globalization [21], clearly accelerated the COVID-19 pandemic. This finding is in line with a report by Coelho et al. [12], which emphasized the role of the air transportation network in this pandemic. In addition, region-specific COVID-19 susceptibility, which was approximated here by BCG vaccination factors, malaria incidence (because COVID-19 susceptibility may be linked to anti-malarial drug use), and the proportion of the population aged over 65 years, explained a substantial part of the variance in COVID-19 case numbers worldwide. This data support the findings by Sala et al. [13] that there is a significant correlation between BCG vaccination and COVID-19 prevalence. Notably, these correlation patterns may change as the pandemic progresses. For example, while the COVID-19 case numbers (per 1 million population) exhibited a relatively robust correlation with malaria incidence, their correlation with the BCG vaccination effect weakened after April 2020, potentially as a result of the recent spread of COVID-19 into more countries with a BCG vaccination program (e.g., Japan, Russia, Turkey, and Brazil).

Our analysis using the regression model, which comprehensively accounted for climate, international human mobility, region-specific COVID-19 susceptibility, and socioeconomic conditions, revealed that climate suitability remains an important driver shaping the current distribution of COVID-19 cases [5, 9]. Although human mobility and host susceptibility to COVID-19 were found to be the main drivers in the spread of COVID-19, the uneven distribution of COVID-19 cases across biome types (Fig 2 and S2 and S4 Videos) suggests that the pandemic may be partially shaped by biogeographical patterns [22]. However, until the pandemic has lasted a full year, it will not be possible to draw reliable conclusions on the relationship between abiotic factors and COVID-19 [7].

Our predictive model does not account for variables relevant to local-scale factors that are associated with community infection or containment/suppression measures implemented against the epidemic in individual countries/regions. Consequently, the model has residuals (Fig 7), i.e., deviations in the observed number of COVID-19 cases that reflect the influence of local-scale drivers on disease spread. Positive deviations in the number of COVID-19 cases may indicate more serious local-scale cluster infections, e.g., in some prefectures in Japan or in parts of South East Asia, Africa, and South America, than predicted by the macro-scale driver-based model, whereas negative deviations in the number of COVID-19 cases indicate the influence of distributional disequilibrium of COVID-19 cases (because SARS-CoV-2 has only recently reached an area, e.g., Africa) or suggest the effectiveness of the present control measures in an area.

There is still a distributional disequilibrium in the global prevalence of infections; the number of confirmed COVID-19 cases changes daily, and the trajectories among countries or regions differ largely (Fig 3 and S1 Fig). The drivers of COVID-19 case numbers indicate a country-specific pattern (Table 1).

The type A countries, with more than 1,000 COVID-19 cases per million in which the infection peaked before mid-April, were mostly the developed countries that had predominant cross-border human mobility in relatively cool and dry climates. The type B countries, with

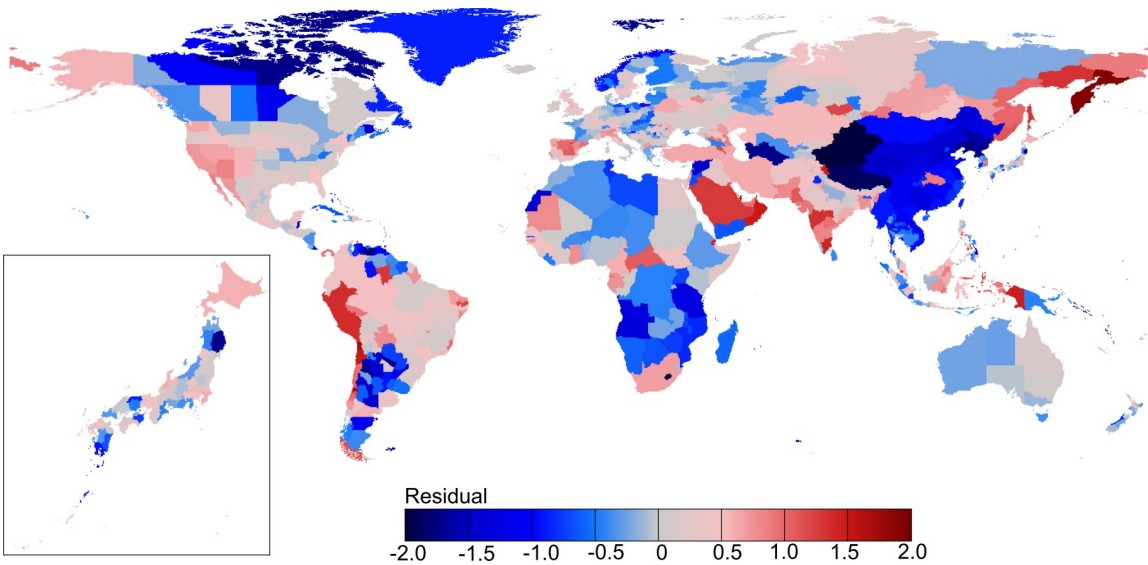

**Fig 7. Residual pattern of the regression model predicting the number of COVID-19 cases (per 1 million population) for 1,020 countries/regions across the globe and for 47 prefectures in Japan.** The map was prepared using shapefile reprinted from a freely available database (GADM; www.gadm.org).

more than 1,000 COVID-19 cases per million in which the infection spread peaked after mid-June, were quasi-developed countries with BCG vaccination programs. The type C countries, with less than 1,000 COVID-19 cases per million in which the infection peaked before mid-April, were countries with high temperature and humidity that are characterized by lower cross-border mobility and more BCG vaccination. The type D countries, with less than 1,000 COVID-19 cases per million in which the infection spread peaked after mid-June, were mostly tropical developing countries with lower population density, less cross-border mobility, higher malaria incidence, and less BCG vaccination. These country-specific factors indicate that the COVID-19 spread is not simply driven by specific environmental variables, and the underlying mechanisms are complicated (Table 1). Therefore, evaluating the drivers of the COVID-19 spread at the present phase of disease expansion is a challenging task.

**Table 1. Drivers of the COVID-19 spread in relation to the country types.** Country types were defined by the patterns of COVID-19 spread (cases per 1 million population) (see Fig 3). Type A, countries that had a peak in the number of COVID-19 cases per week before the middle of April and had more than 1,000 COVID-19 cases per 1 million population; type B, countries that exhibited an increase in the number of COVID-19 cases per week after the middle of June and had more than 1,000 COVID-19 cases per 1 million population; type C, countries that had a peak in the number of COVID-19 cases per week before the middle of April and had less than 1,000 COVID-19 cases per 1 million population; and type D, countries that exhibited an increase in the number of COVID-19 cases per week after the middle of June and had less than 1,000 COVID-19 cases per 1 million population. The statistical significance of differences between the country types was tested by a Bonferroni's multiple comparison test. Different letters indicate the values that are significantly different ($p < 0.05$) from each other.

| Factor | Type A | Type B | Type C | Type D |
|---|---|---|---|---|
| Mean annual temperature | 11.1 (±3.88) [a] | 14.6 (±8.87) [b] | 18.5 (±7.96) [c] | 21.4 (±6.81) [d] |
| Mean annual precipitation | 865 (±368) [a] | 806 (±541) [a] | 1250 (±629) [b] | 1290 (±869) [b] |
| Population density | 485 (±1060) | 342 (±1400) | 391 (±1500) | 164 (±243) |
| Relative frequency of visitors | 154 (±329) [a] | 36.1 (±65.4) [b] | 73.8 (±97.4) [b] | 16.4 (±27.2) [b] |
| GDP per person | 50200 (±21500) [a] | 18500 (±18300) [b] | 22200 (±18100) [b] | 5690 (±5430) [c] |
| BCG vaccination effect | -1.37 (±1.42) [a] | 0.752 (±1.37) [b] | 0.467 (±1.51) [b] | 0.88 (±0.694) [b] |
| Relative frequency of people infected by malaria | 0.163 (±1.26) [a] | 2180 (±14200) [a] | 2950 (±31800) [a] | 40100 (±85000) [b] |
| Relative frequency of people ≥ 65 years old | 18.9 (±3.17) [a] | 11.6 (±4.92) [b] | 14.8 (±6.51) [c] | 7.33 (±4.5) [d] |

The absence of population-wide testing for COVID-19 makes it difficult to investigate the growth dynamics of COVID-19 infection. The case data include a selection bias due to surveillance focusing mainly on symptomatic persons. In particular, the availability of a reverse transcription polymerase chain reaction (PCR) test to identify COVID-19 cases, e.g. the number of PCR tests conducted per population, varies greatly among countries with different medical/public-health conditions (https://ourworldindata.org/covid-testing). Therefore, the true number of the COVID-19 patients and the dynamics of the disease spread are obscured behind the prevalence of asymptomatic carriers [23, 24]. Nevertheless, our findings demonstrate that the COVID-19 pandemic is deterministically driven by climate suitability, cross-border human mobility, and region-specific COVID-19 susceptibility. The present results, based on mapping the spread of COVID-19 and identifying multiple drivers of the outbreak trajectory, contribute to a better understanding of the disease transmission risk and may inform the application of appropriate preventative measures against this pandemic.

## Supporting information

**S1 Fig. The distribution of four country types classified based on the COVID-19 outbreak across biomes.** Type A, countries that had a peak in the number of COVID-19 cases per week before the middle of April and had more than 1,000 COVID-19 cases per 1 million population; type B, countries that exhibited an increase in the number of COVID-19 cases per week after the middle of June and had more than 1,000 COVID-19 cases per 1 million population; type C, countries that had a peak in of the number of COVID-19 cases per week before the middle of April and had less than 1,000 COVID-19 cases per 1 million population; and type D, countries that exhibited an increase in the number of COVID-19 cases per week after the middle of June and had more than 1,000 COVID-19 cases per 1 million population. (TR) tropical rain forest; (TS) tropical seasonal forest/savanna; (TE) temperate rain forest; (SD) subtropical desert; (TD) temperate deciduous forest; (WS) woodland/shrubland; (TG) temperate grassland/desert; (BF) boreal forest; (TU) tundra.
(TIF)

**S2 Fig. Relative importance of explanatory factors in the random forest models explaining the geographical pattern of the cumulative number of COVID-19 cases.** Temp, mean temperature; Prec, mean monthly precipitation; Pop dens, population density; Visitor, relative amount of foreign visitors per population; GDP, gross domestic product per person; BCG, BCG vaccination effect as defined by the first PCA axis summarizing five variables related to BCG vaccination (see the Methods section for details); Malaria, relative malaria incidence; Age, relative proportion of the population aged ≥65 years; First cases, number of days from case onset.
(TIF)

**S3 Fig. Results of additional analyses using the number of conducted COVID-19 tests (sampling effort) as a covariate in the model.** Temp, mean temperature; Prec, mean monthly precipitation; Pop dens, population density; Visitor, relative amount of foreign visitors per population; GDP, gross domestic product per person; BCG, BCG vaccination effect as defined by the first PCA axis summarizing five variables related to BCG vaccination (see the Methods section for details); Malaria, relative malaria incidence; Age, relative proportion of the population aged ≥65 years; First cases, number of days from case onset; Test, number of tests.
(TIF)

**S1 Appendix. List of data sources for the COVID-19 cases numbers.**
(DOCX)

**S1 Video. https://youtu.be/ZIDMtbek-48.**
(TXT)

**S2 Video. https://youtu.be/KlnpUY51D3k.**
(TXT)

**S3 Video. https://youtu.be/UQViOcMFhNk.**
(TXT)

**S4 Video. https://youtu.be/3DpjGoTrk-E.**
(TXT)

## Acknowledgments

We are grateful to the Kubota-lab technical staff for the data management.

## Author Contributions

**Conceptualization:** Yasuhiro Kubota.

**Formal analysis:** Takayuki Shiono, Buntarou Kusumoto.

**Writing – original draft:** Yasuhiro Kubota.

**Writing – review & editing:** Yasuhiro Kubota, Junichi Fujinuma.

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
