## [Decision Letter · Decision Letter 0]

8 Jun 2020

PONE-D-20-14796

Multiple drivers of the COVID-19 spread: role of climate, international mobility, and region-specific conditions

PLOS ONE

Dear Dr. Kubota,

Thank you for submitting your manuscript to PLOS ONE. After careful consideration, we feel that it has merit but does not fully meet PLOS ONE’s publication criteria as it currently stands. Therefore, we invite you to submit a revised version of the manuscript that addresses the points raised during the review process.

Your manuscript was reviewed by 2 experts in the field. Although one reviewer was satisfied with the quality of your work, the other identified a series of important problems that require your attentions. Please review the attached comments and provide your responses.

We look forward to receiving your revised manuscript.

Kind regards,

Yury E Khudyakov, PhD

Academic Editor

PLOS ONE

Journal Requirements:

2. We note that Figure 1 and 6 in your submission contain [map/satellite] images which may be copyrighted. All PLOS content is published under the Creative Commons Attribution License (CC BY 4.0), which means that the manuscript, images, and Supporting Information files will be freely available online, and any third party is permitted to access, download, copy, distribute, and use these materials in any way, even commercially, with proper attribution. For these reasons, we cannot publish previously copyrighted maps or satellite images created using proprietary data, such as Google software (Google Maps, Street View, and Earth). For more information, see our copyright guidelines: http://journals.plos.org/plosone/s/licenses-and-copyright.

a. You may seek permission from the original copyright holder of Figure 1 a to publish the content specifically under the CC BY 4.0 license. 

Reviewers' comments:

Reviewer's Responses to Questions

**Comments to the Author**

1. Is the manuscript technically sound, and do the data support the conclusions?

Reviewer #1: Yes

Reviewer #2: Yes

2. Has the statistical analysis been performed appropriately and rigorously? 

Reviewer #1: Yes

Reviewer #2: Yes

3. Have the authors made all data underlying the findings in their manuscript fully available?

Reviewer #1: Yes

Reviewer #2: Yes

4. Is the manuscript presented in an intelligible fashion and written in standard English?

Reviewer #1: Yes

Reviewer #2: Yes

5. Review Comments to the Author

Reviewer #1: Title: Multiple drivers of the COVID-19 spread: role of climate, international mobility, and region-specific conditions

Authors; Yasuhiro Kubota1*, Takayuki Shiono1, Buntarou Kusumoto2, Junichi Fujinuma1

Manuscript Number: PONE-D-20-14796

Article Type: Research

Reviewer; Mohamed A Daw, MD, PhD, MPS, FTCDI

Major Comments;

I read with great interest this paper and despite a lot of hypothetical assumption and ecological and bio epidemiological factors which is indeed difficult to calculate, I think the paper highlights and inspires a lot of new aspects that to be reflected on the global control of pandemic.

The paper need to be edited and formulated according the guide lines of the Journal-PLOS ONE

DECISION

ACCEPT,

Reviewer #2: The authors evaluated the role of climate, region-specific susceptibility, and international traveller population in shaping the geographical patternsbof COVID-19 cases, results showed that the COVID-19 pandemic is deterministically driven by climate suitability, cross-border human mobility, and region specific susceptibility.

From the perspective of research significance, this research is meaningful, as indicated by authors – “The present results, based on mapping the spread of COVID-19 and identifying multiple drivers of this outbreak trajectory, may contribute to a better understanding of the disease transmission risk and the measures against long-term epidemic”.

However, the novelty of this study research and the full presentation and discussion of the data are slightly inadequate. The comments are listed below:

1. As described by authors - “……several research groups have focused on relevant factors individually and quickly examined the role of climate [8–10], international mobility linked to human contact [11,12], and community-based host susceptibility [13] in the spread of COVID-19”. Compared with published studies, what are the different research findings and novelty of this study? Does it emphasize the relative importance of these factors? If so, are there any interactions between these factors?

2. The “Introduction” is too long, some descriptions such as third paragraph (Evaluating the drivers of the COVID-19 spread is a challenging task at the present phase……) is suggested to be moved to the “Discussion” section.

3. “Results and Discussion”: it was described as “negatively correlated” or “positively correlated” with the accumulated numbers of the COVID-19 cases, but did not make an in-depth discussion on what caused the “negatively correlated” or “positively correlated”. The descriptions showing the correlation are not sufficient enough, the reasons and explanations behind the data are the most important descriptions.

4. “Results and Discussion”: in addition to describing the “negatively correlated” or “positively correlated”, it is suggested to use specific data from some representative countries or regions to explain the relationship between above factors with the accumulated numbers of the COVID-19 cases. Some Tables are suggested to be added to show the specific values.

6. PLOS authors have the option to publish the peer review history of their article (what does this mean?). If published, this will include your full peer review and any attached files.

Reviewer #1: Yes: Mohamed A Daw, MD, PhD, MPS, FTCDI

Reviewer #2: No

---

## [Author Response · Author response to Decision Letter 0]

1 Sep 2020

Editor-in-Chief: Plos One

Dear Sir or Madam,

Thank you very much for your letter on 8 June 2020 to our submitted manuscript. We sincerely appreciate comments from the reviewers. We have carefully revised the manuscript according to the comments and suggestion. Moreover, we updated the data including new case data until 30 June and redid the analysis; therefore, the number of countries/regions slightly changed and the related figures are also revised.

Please find attached our manuscript entitled “Multiple drivers of the COVID-19 spread: the roles of climate, international mobility, and region-specific conditions” to be considered for publication as an article in Plos One.

Our point-to-point responses to the editor’s and reviewers’ comments are given below. We hope that these revisions will result in our paper being acceptable for publication. Thank you very much for your help, and we look forward to hearing your decision on our manuscript.

Yours sincerely,

Yasuhiro Kubota

Reply to the comments by Dr Mohamed A Daw

We appreciate your assessment on our manuscript. We have edited and formulated according the guide lines of the Journal-PLOS ONE.

Reply to the reviewer 2’s comments:

We appreciate constructive comments on our manuscript. Based on the reviewer’s comments, we have conducted additional analyses and revised the manuscript (see the parts written in red). Our point-to-point responses to the comments are listed below.

1. As described by authors - “……several research groups have focused on relevant factors individually and quickly examined the role of climate [8–10], international mobility linked to human contact [11,12], and community-based host susceptibility [13] in the spread of COVID-19”. Compared with published studies, what are the different research findings and novelty of this study? Does it emphasize the relative importance of these factors? If so, are there any interactions between these factors?

In the revision, we clarified novelty of our findings in the abstract (page 2: lines 32-35): “Notably, the relative importance of these factors changed over time; the number of days from outbreak onset drove COVID-19 spread in the early stage, then human mobility accelerated the pandemic, and lastly climate (temperature) propelled the phase following disease expansion”.

2. The “Introduction” is too long, some descriptions such as third paragraph (Evaluating the drivers of the COVID-19 spread is a challenging task at the present phase……) is suggested to be moved to the “Discussion” section.

According to the suggestion, we have revised the introduction section (page 3-5: lines 42-82). With this respect, we also improved the discussion section (page 12-25: see the parts written in red).

3. “Results and Discussion”: it was described as “negatively correlated” or “positively correlated” with the accumulated numbers of the COVID-19 cases, but did not make an in-depth discussion on what caused the “negatively correlated” or “positively correlated”. The descriptions showing the correlation are not sufficient enough, the reasons and explanations behind the data are the most important descriptions.

We added a little interpretation in relation to the correlative patterns (e.g. page 18: lines 306-311). These correlative patterns are changing and the underpinning mechanism is not clear. Therefore, we also stated this point (page 20: 350-354).

4. “Results and Discussion”: in addition to describing the “negatively correlated” or “positively correlated”, it is suggested to use specific data from some representative countries or regions to explain the relationship between above factors with the accumulated numbers of the COVID-19 cases. Some Tables are suggested to be added to show the specific values.

According to the suggestion, we conducted the additional analysis on some representative countries to argue the correlative patterns of environmental factors with the accumulated numbers of the COVID-19 cases. In the revised manuscript, we added new graph (Fig. 3, Fig. S1, and Fig. S2) and Table 1 that show country/region-specific epidemic pattern (page 12-15: lines 204-254, page 21-24: lines 383-418).

---

## [Editor Report · Decision Letter 1]

7 Sep 2020

Multiple drivers of the COVID-19 spread: role of climate, international mobility, and region-specific conditions

PONE-D-20-14796R1

Dear Dr. Kubota,

We’re pleased to inform you that your manuscript has been judged scientifically suitable for publication and will be formally accepted for publication once it meets all outstanding technical requirements.

Kind regards,

Yury E Khudyakov, PhD

Academic Editor

PLOS ONE
---

## [Editor Report · Acceptance letter]

16 Sep 2020

PONE-D-20-14796R1 

Multiple drivers of the COVID-19 spread: the roles of climate, international mobility, and region-specific conditions 

Dear Dr. Kubota:

I'm pleased to inform you that your manuscript has been deemed suitable for publication in PLOS ONE. Congratulations! Your manuscript is now with our production department. 

Kind regards, 

on behalf of

Dr. Yury E Khudyakov 

Academic Editor

PLOS ONE